# Effectiveness of Antimicrobial Lock Therapy for the Treatment of Catheter-Related and Central-Line-Associated Bloodstream Infections in Children: A Single Center Retrospective Study

**DOI:** 10.3390/antibiotics12050800

**Published:** 2023-04-23

**Authors:** Claudia Signorino, Eleonora Fusco, Luisa Galli, Elena Chiappini

**Affiliations:** 1Department of Health Sciences, Meyer Children’s University Hospital IRCCS, University of Florence, Florence 50139, Italy; 2Division of Pediatric Infectious Disease, Department of Health Sciences, Meyer Children’s University Hospital IRCCS, University of Florence, 50139 Florence, Italy

**Keywords:** antimicrobial lock therapy, central venous catheter, catheter-related bloodstream infections, catheter-associated bloodstream infections, children

## Abstract

Antimicrobial lock solutions (ALT) in combination with systemic antibiotics can represent a valid option to attempt central venous catheter (CVC) salvage in the case of catheter-related and central-line-associated bloodstream infections (CRBSI and CLABSI). However, data concerning the effectiveness and safety of ALT in children are limited. We aimed to share our center’s experience in order to contribute to investigations into the causes of ALT failure in the pediatric population. All children consecutively admitted to Meyer Children’s Hospital, University of Florence, Italy, from 1 April 2016 to 30 April 2022, who received salvage ALT to treat an episode of CRBSI/CLABSI, were reviewed. According to ALT failure or success, children were compared with the aim of identifying the risk factors for unsuccessful ALT outcome. Data from 28 children, 37 CLABSI/CRBSI episodes, were included. ALT was associated with clinical and microbiologic success in 67.6% (25/37) of children. No statistically significant differences were observed between the two groups, successes and failures, considering age, gender, reason for use, duration, insertion, type and presence of insertion site infection of the CVC, laboratory data and number of CRBSI episodes. Nevertheless, a trend towards a higher success rate was observed for a dwell time of 24 h for the entire duration of ALT (88%; 22/25 vs. 66.7%; 8/12; *p* = 0.1827), while the use of taurolidine and the infections sustained by MDR bacteria were associated with a tendency toward greater failure (25%; 3/12 vs. 4%; 1/25; *p* = 0.1394; 60%; 6/10 vs. 33.3%; 8/24; *p* = 0.2522). No adverse events, except one CVC occlusion, were observed. ALT combined with systemic antibiotics appears to be an effective and safe strategy for treating children with CLABSI/CRBSI episodes.

## 1. Introduction

Central-line-related and central-line-associated bloodstream infections (CRBSI and CLABSI) are the most common complications of central lines and are associated with an increased risk of mortality, morbidity, and hospital costs [1]. Rates of CLABSI in children vary based on the underlying disease and type of the central venous catheter (CVC) and range from 0.2 to 11.3 infections per 1000 line days [2]. Although catheter removal is considered the traditional preferred management [3], catheter salvage can be attempted, particularly for children who need it for chemotherapy, hemodialysis, and parental nutrition and for whom alternative venous access can be difficult to place [4]. In these cases, systemic antibiotics can be administered in combination with antimicrobial lock solutions (ALT). ALT involves the instillation of a highly concentrated antimicrobial solution, often in association with an anticoagulant, in a volume sufficient to fill the catheter lumen, when it is not in use, in order to eradicate bacteria and fungi embedded in the intraluminal microbial biofilm of the catheter [5]. Recent studies in adults clearly evidenced that the use of antimicrobial lock therapy in conjunction with systemic antibiotics is superior to peripheral antibiotics alone for catheter salvage, showing a success rate greater than 70% [6,7]. Similar results were observed in children, even if the few pediatric studies were small and retrospective; consequently, the efficacy and safety of ALT in this population have not yet been fully clarified [2,4].

In a recent systematic review summarizing results from 661 CLABSI/ CRBSI episodes, higher ALT success (in association with systemic antibiotics) was reported for CRBSI than for CLABSI and for ethanol than for other lock solutions, while high rates of successful catheter salvage were related to coagulase-negative staphylococci infections (CoNS) [4].

Sharing our experience, we aim to provide a contribution to determining the effectiveness of ALT in children, identifying factors affecting its outcome.

## 2. Results

Overall, data from 28 children, 37 CLABSI/CRBSI episodes, were included in the study. The median age for infectious episodes was 58 (IQR 31–116) months. CVC was used for total parental nutrition (TPN) in 13/28 (46.4%), chemotherapy in 11/28 (39.3%), and dialysis in 3/28 (10.7%) patients. In the remaining patient, the CVC was located to administer intravenous immunosuppressive therapy, due to the difficulty in finding peripheral venous accesses. The median duration from catheter insertion to the onset of infection was 250 (IQR 105–427.5) days. Among the 37 episodes, 27 (73.0%) were CLABSI and 10 (27.0%) were CRBSI (Table 1).

The most common isolated bacteria were Gram-negative bacilli (15/37; 40.5%), followed by CoNS (8/37; 21.6%) and *S. aureus* (7/37; 18.9%) (Table 2). CoNS included only *S. epidermidis* and *S. hominis.*

Only one child, with *S. aureus* CLASBI, presented with evidence of secondary dissemination of infection, specifically ankle osteomyelitis. Except for four patients who received taurolidine, CLABSI/CRBSI episodes were treated with a combination of antibiotic lock therapy and systemic antibiotics. The same molecule was administered via CVC and peripheral intravenous catheter (PIVC), even if systemic antibiotic therapy was often multitherapy, in which other antibiotics were associated on the basis of the isolated microorganism.

In the case of a double-lumen catheter, both lumens were locked and the same ALT was injected into them.

In Table 2, the ALT antibiotic/antiseptic and anticoagulant solution concentrations used are reported.

The median time interval from the first day of symptoms to ALT initiation was 5.5 (IQR 3–13) days. The initial dwell time of lock therapy was 24 h in 36/37 (97.3%) episodes. In 6/37 episodes (16.2%), dwell time was reduced to 8–12 h per day, after 3–6 days, in order to use the CVC for systemic antibiotic therapy, TPN or chemotherapy, due to the difficulty of obtaining a peripheral line.

### Outcome Analysis

The ALT success rate was 67.6% (25/37). Reasons for treatment failure were persistent bacteremia (3/12; 25%) or clinical or laboratory findings of infection (3/12; 25%), despite at least 72 h of appropriate lock therapy or recurrence (3/12; 25%), or reinfection (3/12; 25%) within 90 days after the suspension of ALT. In all but one of these cases, the catheter was removed. Other reasons for catheter removal, besides failure, were dislocation (1/37; 2.7%) or end of therapy (2/37; 5.4%). In two episodes, the cause was missing. The pathogens responsible for the lock failure were primarily *Klebsiella pneumoniae* (2/12; 16.7%), CoNS (2/12; 16.7%) and *Candida parapsilosis* (2/12; 16.7%). In 3/12 (25%) failed episodes, taurolidine was the lock solution (Table 2).

Considering complications, occlusion developed in one child (1/37; 2.7%) during taurolidine/4%citrate ALT, but was resolved thanks to thrombolytic drugs.

Characteristics of the two groups of episodes (failure vs. success) are summarized in Table 3.

No significant difference was observed among groups considering age, gender, the reason for use, duration, presence of insertion site infection of the CVC, laboratory data and number of CRBSI episodes.

Although not statistically significant, a trend towards a higher failure rate was observed in episodes treated with taurolidine (25%; 3/12 vs. 4%; 1/25; *p* = 0.1394) and in those sustained by MDR bacteria (60%; 6/10 vs. 33.3%; 8/24; *p =* 0.2522). On the other hand, a dwell time of 24 h for the entire duration of ALT was associated with a trend towards a higher success rate (88%; 22/25 vs. 66.7%; 8/12; *p* = 0.1827). These results were confirmed by the univariate analysis (Appendix A); however, since no statistically significant variables were identified, we did not perform the multivariate analysis.

In four CLABSI/CRBSI episodes which failed regarding the first lock therapy, CVC was not removed but children received a second ALT. In two of them (50%), the second ALT regimen was successful and the CVC was preserved, while in the other two children (50%) the CVC was removed because of re-infection or recurrence.

## 3. Discussion

Currently, evidence for lock therapy in children is insufficient. Studies are limited by the small number of patients or by the study design. Moreover, although in the IDSA guidelines there is a section dedicated to children [3], pediatric practice is mostly influenced by adult studies. Long-term catheters should be removed from patients with CVC infections associated with any of the following conditions: complicated CRBSI; bloodstream infection that continues despite 72 h of antimicrobial therapy to which the infecting microbes are susceptible; infections due to *S. aureus, P. aeruginosa*, fungi, or mycobacteria, and in case of tunnel infections, port abscesses or exit site infections. However, guidelines specify that in the case of CRBSI in children with “unusual extenuating circumstances”, the attempt to salvage the CVC is allowed; therefore, the decision must be assessed on a case-by-case basis [3]. At the same time, there are reports of effective lock therapy in the case of CLABSI/CRBSI due to the pathogens mentioned above [2,4,8].

In our study, for *S. aureus* the rate of successful salvage was 100%. We only treated one episode of *P. aeruginosa* catheter infection with ALT therapy, but it failed. Regarding fungal infections, one out of three cases treated with ALT (33.3%) was successful, supporting the growing idea that catheter salvage can be attempted [9].

In our study, exit site infections were not a criterion for not attempting the catheter salvage, and in this case the ALT success rate was 50%. Furthermore, we reported four children to whom, after the failure of the first lock, selected on the basis of the susceptibility test results, a second one was administered. These were patients who had a crucial need for intravenous access and for whom alternative venous access was difficult to establish. Treatment was carried out with a close monitoring of clinical and laboratory findings, and in addition in this case the outcome was positive for half of them.

In contrast with the current literature, for CoNS CLABSI/CRBSI we detected a quite high initial failure rate (25%), without any episode of recurrence [4].

Although adult studies encourage early catheter removal in the case of CVC Gram-negative infections [10], according to data on the pediatric population we obtained quite favorable results, with a successful ALT rate of 66.7% [4].

Even if it is widely accepted that taurolidine is the most effective lock solution for the prevention of CRBSI/CLABSI in children [11,12], in the Chesshyre et al. review authors specified that it is not recommended as ALT in children, due to limited evidence [13]. Nevertheless, Vassallo et al., in their systematic review, considered taurolidine and ethanol to be two efficacious options for CoNS CLABSI [14], and some adult studies reported promising results [15,16].

In our study, taurolidine was chosen with a therapeutic purpose in four CLABSI/CRBSI episodes, but for three of them it failed. More studies are needed in the pediatric population to define the real efficacy of this antiseptic lock solution, while more positive results have been achieved by ethanol lock therapy [4].

At our center, antibiotic lock therapy was found to have an overall success rate of 67.6% in children with different primary conditions. This value is slightly lower than the previous published data [2,4]. This finding can be partially explained by the small number of patients enrolled and their heterogeneity, the presence of a substantial proportion of oncological children, and the exclusion of neonates, in which a recent study reported a greater success of ALT (success rate of 84.6%) [17].

No statistically significant differences were observed between the two groups of successes and failures. However, a dwell time of 24 h for the entire duration of ALT resulted in a trend toward a higher success rate. The optimal dwell time for ALT is not well defined, even if the majority of clinical studies have suggested that a minimum of 8 h per day, with targets of 12 h per day, is optimal to obtain CVC sterilization [18].

On the other hand, even if not statistically significant, a correlation with a higher failure was noticed in episodes treated with taurolidine and in those sustained by MDR microorganisms. This last finding is in agreement with Freire et al.’s study results, evidencing a higher mortality rate for long-term catheter-related infections due to MDR bacteria [6].

Moreover, in our study, no statistically significant difference in the failure rates was observed between CLABSI and CRBSI episodes. These data are not in line with what was reported by Buonsenso D. et al., who evidenced that ALT plus systemic antibiotic therapy was associated with markedly improved outcomes in children with CRBSI, but not in those with CLABSI, probably due to the fact that in the latter the source of infection may not be the catheter itself [4].

Considering the risks of ALT, occlusion can occur. We observed occlusion in one patient during taurolidine/4%citrate ALT. In the literature, it is reported that TauroLock can lead to catheter clotting [19]. Nevertheless, we did not include this patient in the failure group, since the occlusion was resolved with the use of thrombolytics and CVC was preserved.

This study has several limitations, so our results and conclusions must be interpreted with caution. These include a small cohort, retrospective data, a single-center design, diverse underlying conditions, a wide variability of lock solutions used and microorganisms isolated, and the absence of a comparison with a group of patients treated with systemic antibiotics only, since in our hospital no child receives this kind of approach.

However, although the data in our report were limited to 37 episodes of infection, most of the pediatric studies in the literature are of lower or similar size [20,21,22] and information on this argument is scarce. Therefore, it is essential to bring to light real-life experiences such as ours.

In conclusion, even though ALT in addition to systemic antibiotics appears to be an efficacious and safe strategy for treating children with CRBSI or CLABSI episodes, larger, multi-center studies, especially on randomized controlled trials, are required to support the currently available literature data.

## 4. Materials and Methods

### 4.1. Study Design

We performed a retrospective study, including all children consecutively admitted to Meyer Children’s Hospital, University of Florence, Italy, from 1 April 2016 to 30 April 2022 who received salvage ALT to treat an episode of CLABSI/CRBSI.

Inclusion criteria were: age between 1 month and 18 years; the presence of a central venous catheter; signs or symptoms of infection (e.g., fever, chilis) and no apparent clinical origin other than the catheter; possible presence of secondary disseminations of infection; bacteremia confirmed by at least one blood culture taken through a peripheral vein puncture or through the catheter (for Coagulase-negative Staphylococci, CoNS, the growth of at least two blood cultures taken at different times was required); administration of at least 72 h of lock therapy.

All isolated pathogens were included.

Some patients were included more than once, but only if the second CLABSI/CRBSI episode occurred at least 90 days after the end of treatment of the previous one. Otherwise, reinfection or recurrence was considered.

Exclusion criteria were: age <1 month or >18 years; absence of a CVC; negative blood culture; an alternative source of bloodstream infection (BSI); non-administration of lock therapy or lock therapy duration less than 72 h.

Data of eligible patients were collected and entered into an electronic database and analyzed.

Patients were determined to have two possible outcomes: ALT failure or success.

ALT failure consisted of a lack of clinical or laboratory improvement (return to normal levels or in any case substantial improvement of inflammatory markers), or persistent bacteremia after at least 72 h of appropriate therapy (systemic and lock antimicrobial therapy to which the isolated microorganism was susceptible), followed by the CVC removal, or recurrence or reinfection within 90 days after the suspension of ALT.

For patients who received one more lock therapy after the failure of the first one, we evaluated the outcomes in terms of the success or the failure of the second treatment.

We did not include the removal of the catheter for reasons unrelated to the infection among the criteria of failure.

The study was approved by the Ethics Committee of the Meyer Children’s Hospital. According to age, tutors or patients gave written informed consent before the beginning of the study.

### 4.2. Definitions

Based on the US guidelines definitions, CVC infections were categorized as follows:

CLABSI: defined as the recovery of a pathogen from blood culture (a single blood culture for an organism not commonly present on the skin, and two or more blood cultures for an organism commonly present on the skin) in a patient who had a central line at the time of infection or within 48 h before the development of infection. The infection cannot be related to any other infection the patient might have [23].

CRBSI: definite diagnosis requires one of the following: isolation of the same pathogen from a quantitative blood culture drawn through the central line and from a peripheral vein with the single bacterial colony count at least threefold higher in the sample from the central line as compared to that obtained from a peripheral vein; (or) same organism recovered from percutaneous blood culture and from quantitative (>15 colony-forming units) culture of the catheter tip; (or) a shorter time to positive culture (>2 h earlier) in the central line sample than the peripheral sample [3].

Exit site infection: signs of inflammation confined to an area (typically < 2 cm) surrounding the catheter exit site and the presence of exudate that proved to be culture positive [24].

Multi drug resistance (MDR) isolated pathogen: acquired nonsusceptibility to at least one agent in 3 or more antimicrobial categories for Gram negative and positive, except Staphylococcus; resistance to one key antimicrobial agent (methicillin) only for Staphylococcal strains [25].

Persistent bacteremia: defined as bacteremia that persists despite at least 72 h of antimicrobial therapy selected on the basis of the antibiogram [3].

Recurrence: it was established when a new CLABSI or CRBSI episode caused by the same organism, with a comparable antibiogram, was diagnosed less than 90 days following the completion of successful treatment of previous BSI, with catheter salvage. Recurrence data beyond 90 days were not included because they are considered unlikely to be substantially influenced by the primary treatment [7].

Reinfection: it has been defined as an infection from a different microorganism, diagnosed less than 90 days following the completion of successful treatment of previous BSI, with catheter salvage.

### 4.3. Lock Therapy Procedure

In our hospital, the decision to start ALT with concomitant systemic antimicrobials for the treatment of CVC infection is made, case by case, by the physicians taking care of the children, based on the current literature data [14,26]. In particular a multidisciplinary team, always involving a pediatric infectious disease specialist expert in the treatment of CLABSI/CRBSI, usually decides the type of solution (antiseptic or antibiotic), drug concentration and eventual use of heparin, considering the patient’s history, clinical status, the causative infectious organism and the drug susceptibility test results (if available). The latter was carried out by a reference method, broth micro-dilution [27], and was interpreted according to EUCAST clinical breakpoints [28] both for bacteria and fungi. ALT is always replaced by aspirating the solution already present in the CVC lumen, without flushing. All solutions administered were known to be stable for ALT [14,26].

During the treatment period, serial blood cultures from CVC were performed to monitor lock efficacy. ALT therapy was performed as a standard of care therapy; therefore, parents/tutors were not asked to sign written consent to ALT, but only verbal informed consent was obtained.

### 4.4. Statistical Analysis

The continuous variables were expressed as the median and interquartile range (IQR). The categorical variables have been expressed as numbers and percentages.

According to ALT outcome (failure or success) children were categorized into two groups and were compared on the basis of age, gender, the reason for use, duration and eventual presence of insertion site infection of the CVC, laboratory data, number of CRBSI episodes, presence of MDR isolated pathogen, number of days from the onset infection to the start of lock therapy, type of ALT (taurolidine vs. antibiotic) and dwell time (distinguishing those with 24 h for all lock therapy duration and those with different schemes). For the categorical variables, the χ2 or Fisher test was performed, as appropriate; for the continuous variables, the medians between groups were compared using the Wilcoxon Mann–Whitney U test. *p* < 0.05 was considered statistically significant. Univariate logistic regression analysis was performed to investigate the risk factors for ALT failure in the pediatric population. All variables that resulted from significant univariate analyses were included in the multivariate model.

Statistical analyses were performed using the STATA/SE version 11 software package (Stata Corporation, College Station, TX, USA).

## Figures and Tables

**Table 1 antibiotics-12-00800-t001:** Demographic and clinical features of 28 children with 37 BSIs episodes.

Characteristics of 28 Children with 37 BSIs Episodes	Values
Age, m, median (IQR) *	58 (31–116)
Gender, *n* (%)	
Male	20 (71.4)
Female	8 (28.6)
Reasons for using CVC, *n* (%)	
TPN	13 (46.4)
Chemotherapy	11 (39.3)
Dialysis	3 (10.7)
Other **	1 (3.6)
CVC insertion, *n* (%) *	
Innominate vein	18 (48.7)
Internal jugular vein	3 (8.1)
Subclavian vein	6 (16.2)
Brachiocephalic veinMissing	2 (5.4)8 (21.6)
Catheter type, *n* (%) *	
Port-A	5 (13.5)
Tunneled single-lumen catheter	23 (62.2)
Tunneled double-lumen catheter	9 (24.3)
WBC cell/mm3, at the onset of infection, median (IQR) *	5635 (3852.5–11,132.5)
WBC cell/mm3, *n* (%) *	
<11,000	27 (73.0)
≥11,000	10 (27.0)
ANC at the onset of infection, median (IQR) *	4139 (1654–8446.5)
ANC cell/mm^3^, *n* (%) *	
≤1500	9 (24.3)
1500–7000	17 (46)
≥7000	11(29.7)
CRP, mg/dl, at the onset of infection, median (IQR) *	4.725 (1.345–9.1)
CPR mg/dl, *n* (%) *	
≤2	14 (37.8)
2–7	11 (29.7)
≥7	12 (32.4)
Procalcitonin, ng/mL, at the onset of infection, median (IQR) *	8.26 (1.9–43.85)
Procalcitonin ng/mL, *n* (%) *	
≤2	9 (24.3)
2–10	11 (29.7)
≥10Missing	13 (35.1)4 (10.9)
Type of infection, *n* (%) *	
CRBSI	10 (27.0)
CLABSI	27 (73.0)
Duration from catheter insertion to the onset of infection, days, median (IQR) *	250 (105–427.5)
Time to lock therapy, days, median (IQR) *	5.5 (3–13)
Catheter removal, *n* (%) *	16 (43.2)
Successful ALT in CRBSI/CLABSI, *n* (%) *	28 (75.7)

* Percentage and IQR were calculated over 37 episodes. ** One giant cell hepatitis for receiving immunotherapy because of difficulty in finding vascular accesses. IQR: interquartile range; CVC: central venous catheter; TPN: total parental nutrition; WBC: white blood cells; ANC: absolute neutrophil count; CRP: C- reactive protein; CRBSI: catheter-related bloodstream infection; CLABSI: catheter-associated bloodstream infection; ALT: antimicrobial lock therapy; m: months.

**Table 2 antibiotics-12-00800-t002:** Microbiological data, patient characteristics, treatment, and outcomes for 37 BSIs episodes (failures are in red).

	Microorganism	CLABSI /CRBSI *n* (%)	Age, m	Primary Condition	*N*° of Episode	Catheter Insertion Site Infection	MDR	ALT	MIC μg/mL	Antimicrobial Concentration (mg/mL)	Anticoagulant Concentration (IU/mL)	Time to ALT, d	Dwell Time, h	Duration, d	I day of Negative Culture Since The Start of ALT	Persistent Bacteremia/Clinical or Laboratory Infection Findings Despite 72 H of ALT	Recurrence	Catheter Salvage
Gram +	Coagulase-	8	27	PID	I	No	Yes	Daptomycin	1	5	4000	28	24	23	3	No	No	Yes
19 (51.3%)	negative Staphylococci	(21.6%)	87	CRF	I	No	No	Teicoplanin	≤0.12	10	2500	5	24 for 2 d,	7	7	No	No	Yes
													then 12					
			65	Thalassemia	I	No	Yes	Daptomycin	NA	5	4000	3	23	11	8	No	No	Yes
			214	OS	I	No	Yes	Daptomycin	0.5	5	2000	14	24	11	5	No	No	Yes
			210	GC	I	No	Yes	Taurolidine	NA	2%	NA	34	24 for 3	12	5	No	Yes	Yes
													d, then 12					
			167	SBS	I	No	No	Cefazolin	NA	5	2500	20	24	4	9	No	No	Yes
			48	ALL	I	No	Yes	Vancomycin	1	5	2500	13	24	11	11	Yes	NA	No
			208	ALM	I	No	Yes	Vancomycin	1	5	2500	3	24	5	5	No	NA	No (end of therapy)
	*S. aureus*	7	56	CRF	I	No	No	Vancomycin	1	16.6	NA	4	24	14	6	No	No	Yes
		(18.9%)	119	CRF	III	No	No	Taurolidine	NA	2%	Citrate	3	24	3	3	No	No	Yes
											4%							
			123	CRF	IV	No	No	Daptomycin	0.5	5	5000	2	24	16	3	No	No	Yes
			131	CRF	V	No	No	Daptomycin	0.5	5	5000	2	24	20	5	No	NA	No (dislocation)
			16	CCD	I	Yes	No	Teicoplanin	≤0.5	10	2500	2	24	7	3	No	NA	No
			165	CRF	I	Yes	No	Vancomycin	1	11.1	3333.3	5	24	14	9	No	No	Yes
			47	Thalassemia	I	No	No	Vancomycin	1	5	2500	3	24	4	3	No	NA	No (end of therapy)
	*Bacillus* *cereus*	1 (2.7%)	58	ALL	I	No	No	Levofloxacin	0.12	5	NA	1	24	10	NA	Yes	NA	No
	*Bacillus thurigiensis*	1 (2.7%)	65	ALL	I	No	No	Vancomycin	2	5	250	13	24	10	4	No	NA	No (reinfection)
	*E. faecalis*	1 (2.7%)	18	CTE	I	No	No	Teicoplanin	≤0.5	10	2500	4	24 for 3 d then 12	8	6	No	NA	No (reinfection)
	*Enterococcus* *hirae*	1 (2.7%)	21	SBS	I	No	No	Vancomycin	≤0.5	2.5	2500	4	24 for 6	14	5	No	No	Yes
													d, then 12					
Fungi	*C. parapsilosis*	3(8.1%)	110	CRF	II	No	NA	Taurolidine	NA	2%	Citrate	8	24	3	NA	Yes	NA	No
3 (8.1%)											4%							
			159	CF	I	No	NA	Caspofungin	0.12	3	NA	9	24	10	4	No	Yes	No
			73	Syndrome	I	No	NA	Caspofungin	0.25	2	NA	5	24	9	5	No	No	Yes
Gram -	*K.*	6	34	CTE	II	No	No	Ceftazidime	≤0.25	5	2500	12	24	9	6	No	No	Yes
15	*pneumoniae*	(16.2%)	3	Syndrome	I	No	Yes	Imipenem	≤0.12	NA	NA	3	24	NA	14	No	No	Yes
(40.5%)			61	SBS	I	No	Yes	Amikacin	≤4	2	NA	23	24	10	6	No	No	Yes
			28	SBS	I	No	Yes	Tigecycline	0.5	10	NA	2	24 for 5 d, then 8	11	6	No	Yes	No
			50	ALL	I	No	Yes	Gentamicin	≤1	1	2500	9	24	10	6	No	NA	No
			46	PID	II	Yes	Yes	Taurolidine	NA	NA	NA	15	24	16	NA	Yes	NA	No
	*E. coli*	3 (8.1%)	113	SBS	I	No	Yes	Gentamicin	≤1	1	2500	2	24	20	NA	Yes	NA	No
			25	GCH	I	No	No	Ceftazidime	≤0.12	5	2500	10	24	8	4	No	No	Yes
			30	IBD	I	No	NA	Gentamicin	NA	1	2500	3	24	15	5	No	No	Yes
	*Morganella*	1 (2.7%)	42	CTE	III	No	Yes	Ciprofloxacin	≤0.06	0.2	4500	11	24	10	5	No	No	Yes
	*morganii*																	
	*S.*	2 (5.4%)	73	RMS	I	No	No	Levofloxacin	≤1	5	NA	6	24	10	6	No	No	Yes
	*maltophilia*																	
			39	NB	I	No	No	Levofloxacin	≤2	5	NA	4	24	10	4	No	No	Yes
	*E. cloacae*	1 (2.7%)	10	SBS	II	No	Yes	Tigecycline	1	10	NA	8	24 for 3 d then 12	14	5	No	NA	No (reinfection)
	*Pseudomonas putida*	1 (2.7%)	85	RMS	II	No	No	Gentamicin	≤4	1.2	2500	3	24	9	3	No	No	Yes
	*Pseudomonas aeruginosa*	1 (2.7%)	32	HCL	I	Yes	No	Ciprofloxacin	0.25	0.2	4500	2	24	4	NA	Yes	NA	No

CRBSI: catheter-related bloodstream infection; CLABSI: catheter-associated bloodstream infection; PID: primary immunodeficiency; CRF: chronic renal failure; OS: Osteosarcoma; GC: gastric cancer; SBS: short bowel syndrome; ALL: acute lymphoid leukemia; ALM: acute myeloid leukemia; CCD: congenital chloride diarrhea; CTE: congenital tufting enteropathy; CF: cystic fibrosis; GCH: giant cell hepatitis; IBD: inflammatory bowel disease; RMS: rhabdomyosarcoma; NB: neuroblastoma; HCL: Langerhans-cell histiocytosis; MDR: multi drug resistance; NA: not available; ALT: antimicrobial lock therapy; MIC: minimal inhibitory concentration; h: hours; m: months; d: days.

**Table 3 antibiotics-12-00800-t003:** Comparison of patient groups with ALT success and failure.

	Success (*n* = 25)	Failure (*n* = 12)	*p*
Age, m, (IQR)	61 (34–119)	53 (31–110.75)	0.74896
Gender, *n* (%)			
Male	16 (64)	9 (75)	0.711
Reason for using CVC, *n* (%)			
TPN	10(40)	6 (50)	0.7258
Chemotherapy	8 (32)	5 (41.7)	0.7161
Dialysis	6 (24)	1 (8.3)	0.3891
Other	1 (4)	0	1
ANC/ mm3 (IQR)	3864 (1241–10,103)	3996 (2943.5–4794)	0.72634
CRP, mg/dl (IQR)	4.04 (1.34–7.4)	5.68 (1.6225–9.205)	0.32218
Procalcitonin, ng/mL (IQR)	8 (2.1–79.3)	3.7 (1.6–42.35)	0.64552
Catheter days, (IQR)	234 (97.5–376.5)	270 (135–420)	0.60306
CRBSI, *n* (%)	6 (24)	4 (33.3)	0.7004
MDR, *n* (%)	8/24 (33.3) + 1 *Candida parapsilosis*	6/10 (60) + 2 *Candida parapsilosis*	0.2522
Time to ALT, d (IQR)	4 (3–6)	8 (2–11)	0.87288
Taurolidine, *n* (%)	1 (4)	3 (25)	0.1394
CVC insertion site infection, *n* (%)	3 (12)	2 (16.7)	1
Dwell time of 24 h, *n* (%)	22 (88)	8 (66.7)	0.1827

IQR: interquartile range; CVC: central venous catheter; TPN: total parental nutrition; ANC: absolute neutrophil count; CRP: C-reactive protein; CRBSI: catheter-related bloodstream infection; MDR: multi drug resistance; ALT: antimicrobial lock therapy; h: hours; m: months; d: day.

## Data Availability

Data are contained within the article.

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
