# Peer review of "Effectiveness of Antimicrobial Lock Therapy for the Treatment of Catheter-Related and Central-Line-Associated Bloodstream Infections in Children: A Single Center Retrospective Study"

_antibiotics, 2023, doi:10.3390/antibiotics12050800_

Round 1

Reviewer 1 Report

This is a retrospective study performed in the pediatric population in a hospital in Italy on the use of antibiotics lock therapy (ALT) for the treatment of catheter-related bloodstream infections. The authors summarized their experience and based on the available data performed further statistical analysis in order to identify risk factors that predict a high failure rate of ALT. Some of the parts of this manuscript were well written, however, there are some areas that I would like to point out some important areas that the authors should improve before publication.

Major comments:

1. Number of errors were found when going through the tables of this paper. The problem is especially significant in Table 2. The authors should consider further processing Table 2 for readers to visualize your results easily. To be more specific:

Table 1:

a) The row of ANC, the unit /mm3 should go to the title but not the individual row and there is probably an error in the calculation of % as they do not add up to 100%.

b)  The unit is missing for "Time to lock therapy".

c) "CVC insertion": The calculated % does not add up to 100%. The authors may need to reconsider whether they should use the 28 children or the 37 BSI episodes as the base for calculation.

Table 2:

a) The table is too difficult to visualize as there are too many columns in this table. It may be better to use a red font instead of highlighting the whole row in red to indicate treatment failure.

b) The unit of MIC is missing in the table. It would be better for the authors to use instead of <= in the table to reduce confusion. Furthermore, MIC should read as 0.12 instead of 0,12. 

c) Name of the bacteria in the table should be italicized 

d) In one row of coagulase-negative Staphylococcus, data is missing for "catheter salvage".

e) In the row of Enterococcus hirae, the MIC of Vancomycin shown was < or = 0.55, this is likely to be a typo as usually in the microbiology laboratory, Vancomycin MIC is tested at the level of 0.5 instead of 0.55.

f) Please check for any spelling errors in this table, e.g. "Caspofungin", and "Tigecycline".

2. The authors have included minimal inhibitory concentration (MIC) data in their analysis. It may be better for the authors to include the method of performing MIC testing for bacteria and fungus in their methodology session.

3. The authors have included coagulase-negative Staphylococcus in the analysis. It would be good if the authors could provide the exact species of Staphylococcus or comment on whether the coagulase-negative Staphylococcus included were Staphylococcus lugdunensis. This is because Staphylococcus lugdunensis although considered a coagulase-negative Staphylococcus, its virulence is similar to Staphylococcus aureus, and should be treated as such.

4. The definition of MDR used in this study is "acquired nonsusceptibility to at least one agent in 3 or more antimicrobial categories". I do not think this is an appropriate definition for this study. It depends on how many antimicrobial classes your microbiology laboratory tested for in each of the organisms. For some of the organisms for example Stenotrophomonas maltophilia and Enterococcus, your laboratory may test for fewer than 3 classes of antimicrobial classes (depending on whether your laboratory follows CLSI or EUCAST). The definition used is more applicable to cases like Pseudomonas aeruginosa,  Acinetobacter baumanii +/- Enterobacteriaceae, but not to Gram-positive organisms. A more reasonable approach would be to consider methicillin-resistant Staphylococcus as drug-resistant bacteria in your analysis. Authors may rethink how to better define "MDR" in their study.

Minor comments:

1. All bacteria names should be italicized and spell-checked. Line 109 Klebsiella pneumoniae instead of Klebsiella pneumonia. Furthermore, when italicizing bacteria names, watch out for italicizing other text during word processing e.g. Line 110 "and" is being italicized.

2. The authors include Table 4 on patients receiving more than one ALT. I am not sure whether there is much significance in including the table here. The authors may want to elaborate on that.

3. There are several formatting errors identified during the review, e.g. reference  [1,2] instead of [1],[2], Line 150-152, referencing style is also not in the format of MDPI journals.

4. I noticed a wide range of antibiotics was used for antibiotic lock therapy, including the use of beta-lactam, tigecycline, and quinolones. As mentioned in the methodology part, the decision was usually made by the parent team of infectious disease specialists. I am actually amazed by the range of antibiotics that the hospital used, but I am also worried whether the stability of some of the antibiotics such as beta-lactam and tigecycline is one of the reasons for the higher failure rate as mentioned by the authors. Antibiotics such as amikacin, gentamicin, and vancomycin, are commonly used in topical therapies (e.g. antibiotics lock, cement spacer in orthopedics infection) due to their excellent stability in the environment. I am unsure whether other groups of antibiotics as mentioned have good stability to be used for antibiotics lock therapy. Authors may supplement and elaborate on this matter.

Overall I think this paper is a good paper summarizing the experience of ALT in children in the hospital. However, given the number of numerical errors identified during the review process, and many formatting +/- spelling errors (including names of bacteria) when receiving this version of the manuscript, I would recommend major revision.

Reviewer 2 Report

Reviewer comment in the file attached.

Round 2

Reviewer 1 Report

Thank you for the response from the authors. They addressed the raised issues during my last review. For the big table, it may be better presented using the landscape orientation instead of portrait orientation, but I believe this is up to the decision of the journal instead of the authors. The only comment remaining (the reason for recommending "accept after minor revision") is for the authors to recheck all the bacteria names again, as there are still occasional errors seen in the revised manuscript, examples:

- coagulase-negative Staphylococcci instead of coagulase-negative Staphylococci (coagulase-negative Staphylococci refers to a group of bacteria but not a particular bacteria name, therefore it should not be italicized) - Table 2

- Klebsiella pneumoniae instead of Klebsiella Pneumoniae - Table 2

- Line 143: S. aureus, P. aeruginosa should be italicized

Author Response

Thank you for your review. We re-read our manuscript carefully and corrected some errors related to bacteria names.